# Permeability and Toxicity of Cryoprotective Agents in Silkworm Embryos: Impact on Cryopreservation

**DOI:** 10.3390/ijms252111396

**Published:** 2024-10-23

**Authors:** David Urbán-Duarte, Shuichiro Tomita, Hiroki Sakai, Hideki Sezutsu, Horacio Álvarez-Gallardo, Yooichi Kainoh, Seiichi Furukawa, Keiro Uchino

**Affiliations:** 1Centro Nacional de Recursos Genéticos, Instituto Nacional de Investigaciones Forestales, Agrícolas y Pecuarias, Tepatitlán de Morelos, Jalisco 47600, Mexico; alvarez.horacio@inifap.gob.mx; 2Institute of Agrobiological Sciences, National Agriculture and Food Research Organization, 1-2 Owashi, Tsukuba 305-8634, Japan; tomita@affrc.go.jp (S.T.); sakaih786@affrc.go.jp (H.S.); hsezutsu@affrc.go.jp (H.S.); 3Faculty of Life and Environmental Sciences, University of Tsukuba, Tennodai 1-1-1, Tsukuba 305-8572, Japan; kainoh.yooichi.gf@u.tsukuba.ac.jp (Y.K.); furukawa.seiichi.ew@u.tsukuba.ac.jp (S.F.)

**Keywords:** silkworm, embryo, cryoprotective agent, vitrification, *Bombyx mori*

## Abstract

The permeation of cryoprotectants into insect embryos is critical for successful cryopreservation. However, the permeability of silkworm embryos to cryoprotectants and the effects of cryopreservation remain poorly studied. In this study, we evaluated the permeability and toxicity of four cryoprotective agents (CPAs) as well as the vitrification effect on the viability of silkworm embryos. Among the four CPAs, propylene glycol (PG) showed the best permeability. Ethylene glycol (EG) and PG were the least toxic CPAs, but glycerol (GLY) and dimethyl sulfoxide (DMSO) were more toxic. Moreover, we examined several factors including the kind and the concentration of CPAs, exposure time, embryonic stage, and silkworm strains. Embryos at the earlier phases of stage 25 were more tolerant to vitrification using EG. We found that over 21% of embryos treated with EG at the early 2 phase of stage 25: 163 h after egg laying (AEL) developed and progressed to serosa ingestion after vitrification and rewarming. The result was the same in other strains as well. Our results are valuable for the development of new cryopreservation protocols of silkworm embryos.

## 1. Introduction

The survival of cells and organisms during cryopreservation is critically dependent on the prevention of ice crystallization by intracellular CPA [1]. A number of CPAs, including DMSO, EG, GLY, and PG, have been proven effective for the cryopreservation of cells and embryos [2,3,4]. Each CPA has unique biological and biophysical properties, and the protective effects of CPAs are generally considered colligative [5]. It is, therefore, important to select CPAs with low cytotoxicity and high permeability [6,7]. If the permeation is too slow, the cells must be exposed to a CPA for an extended period, which may result in the cell being subjected to the chemical toxicity of the CPA [6]. Additionally, the viability of cryopreserved embryos is also affected by several factors, including the type of CPA, exposure time, temperature, cooling and thawing rate, and embryonic development [8,9,10,11,12]. Consequently, it is essential to optimize for each species.

Successful cryopreservation of some dipteran and lepidopteran embryos has been achieved through the development of vitrification protocols [8,9,10]. Vitrification is defined as the solidification of a liquid into a non-crystalline or amorphous solid known as glass [13]. The initial step in vitrification of insect embryos is the introduction of CPA into the embryo. In previous studies, we developed an easy and effective protocol for dechorionation and permeabilization on silkworm embryos [14,15], which rendered the silkworm embryos permeable to both the water and the CPA. Moreover, our research revealed that an embryonic stage [early 1 phase-stage 25, appearance of taenidium, 163 h AEL] exhibits high permeability and viability following permeabilization [15]. Furthermore, Fukumori et al. [11] also reported a higher tolerance to vitrification at stages between Stages 24 and 25 of silkworm embryos when EG was used as the CPA. However, there is currently no information available regarding the toxicity and permeability of different CPAs such as DMSO, GLY, and PG in silkworm embryos. The present study aimed to examine the permeability and toxicity of four CPAs, as well as on the effect after vitrification and rewarming. The findings demonstrated that the toxicity of EG and PG was less pronounced than that of the other CPAs. PG exhibited the most effective permeation into the embryo. The vitrificated and rewarmed embryos using EG could develop and progress to serosa ingestion. The results obtained here are valuable for the development of embryo cryopreservation protocols for other insects.

## 2. Results

### 2.1. Osmotic Responses to Different CPA Solutions

We first investigated the osmotic responses to CPAs. The osmotic responses to DMSO, EG, GLY, and EG solutions at different concentrations (0.5, 1, 2, and 4 M) for 120 min at 25 °C were studied by measuring and analyzing the relative area changes of the embryos to CPAs. We used the silkworm eggs in early 1 phase-stage 25, which we found in a previous report with better permeability [15]. We commonly observed initial shrinkage and subsequent gradual re-expansion in all the CPAs (Figure 1). There were large differences between types and concentrations of CPAs on the minimum relative area reached and the re-expansion speed of the embryos (Figure 1). At higher CPA concentrations, the relative area decreased and reached the minimum relative area faster, and similarly, the re-expansion was faster. However, the re-expansions did not return to the initial relative area of embryos before CPA treatment, except for 4 M PG (Figure 1).

On the other hand, when the embryos were exposed to each 2 M solution of DMSO, EG, GLY, and PG, significant differences were observed (Figure 2); the degree and speed of shrinkage and re-expansion were relatively different among them. The highest minimal relative area was observed in PG at 20 min (89.8%), followed by DMSO at 25 min (79.7%), EG at 30 min (72.0%), and GLY at 50 min (67.3%) as shown in Figure 2. Embryos exhibiting the higher re-expansion at 120 min were found in PG (96.3%), followed by DMSO (88.7%) and EG (82.2%). In particular, the re-expansion of the embryos in GLY (69.13%) was extremely low compared to the other CPAs (Figure 2).

### 2.2. Effect of Different CPA Solutions on the Embryonic Viability

To investigate the viability of embryos, we examined the effects of exposure to 2 M DMSO, EG, GLY, and PG solutions for 30 min in embryos at the early 1 phase-stage 25. There were no significant differences in the proportion of developing embryos among the CPAs. Treatment with 2 M GLY significantly decreased the rate of hatching (z = 3.606, *p* = 0.000) compared to untreated permeabilized embryos as shown in Figure 3A. Embryos treated with EG and PG developed to the second instar, although we observed a decrease compared to embryos not exposed to CPAs: control. No embryos developed to the second instar with the DMSO and GLY treatments (Figure 3A). The treatment with DMSO and GLY tended to delay the development of the embryos; this was more remarkable in GLY (Figure 3B).

### 2.3. Effect of Different Exposure Times of 2 M EG and PG Solutions on the Embryonic Viability

As shown in Figure 3, the 2 M EG and PG treatments appeared less toxic than the DMSO and GLY treatments. In addition, the results in Section 2.1 show that the re-expansion of the embryos treated with 2 M EG and PG treatments was slow and did not regain their original area even after 120 min. We focused our efforts on the tolerance of embryos in the early 1 phase-stage 25 to EG and PG, and different exposure times. The exposure of embryos to 2 M EG revealed no significant differences in the proportion of developing embryos at 30, 60, and 90 min in comparison with the control. However, at 120 min, a significant decrease was observed (*z* = 3.602, *p* = 0.000) (Figure 4). A similar significant decrease was observed in the hatching and the embryos developed to the second instar at 30 min of exposure (Figure 4). In contrast, at 60, 90, and 120 min of exposure, the hatching rate (60 min: *z* = 2.061, *p* = 0.039; 90 min: *z* = 2.631, *p* = 0.008; 120 min: *z* = 3.881, *p* = 0.000) decreased significantly (Figure 4). The longer exposure times of 60, 90, and 120 min had a much larger detrimental effect on the embryos that developed to the second instar (60 min: *z* = 2.561, *p* = 0.010; 90 min: *z* = 2.428, *p* = 0.015). There were no embryos developed to the second instar at 120 min (Figure 4). On the other hand, the 2 M PG treatment exhibited no notable differences in the proportions of developing embryos and hatching at 30, 60, and 90 min in comparison to the control. However, the proportion of embryos developed to the second instar larvae dramatically decreased at 90 min (90 min: *z* = 3.036, *p* = 0.002). The longest exposure time (120 min) had a more detrimental effect (Developing embryos: *z* = 3.071, *p* = 0.002; Hatching: *z* = 2.668, *p* = 0.007), with no embryos developing to the second instar (Figure 4).

During the process of removing the EG and PG with a sequential trehalose solution (0.5, 0.25, 0.125 M) and Grace’s medium, it was observed that some of the embryos treated for 60, 90, and 120 min in a 2 M EG solution were damaged and some were not (Figure 5A,B). Similar findings were observed in the embryos treated for 90 and 120 min in 2 M PG solution (Figure 5C,D). This damage tended to occur in the Grace’s medium step, which involved two 5 min treatments with Grace’s medium to remove the CPAs.

### 2.4. Sensitivities of Embryos to Vitrification Solutions

The effect of different vitrification solutions on embryos was evaluated. The permeabilized embryos in early 1 phase-stage 25 were subjected to two vitrification solutions: (1) 2 M EG solution for 30 min followed by vitrification solution of 7.2 M EG with 0.5 M trehalose for 5, 10, 20 min; (2) 2 M PG solution for 30 min followed by vitrification solution of 5.5 M PG with 0.5 M trehalose for 5, 10, 20 min. The effects of vitrification solution without immersion in liquid nitrogen are shown in Figure 6. In the case of EG, no significant differences to the control were found in the proportions of developing embryos despite varying exposure times. However, the hatching rate was significantly reduced in the 20 min exposure period (*z* = 2.742, *p* = 0.006), and the growth of the larvae to the second instar was markedly reduced across all exposure times (Figure 6). On the other hand, there were no significant differences to the controls in PG in the proportion of developing and hatching of embryos in spite of different exposure times. However, the proportion of embryos developing to the second instar decreased in all exposure times; the 10 min exposure had a significant detrimental effect (*z* = 2.318, *p* = 0.020), and no embryos developed to the second instar in the 20 min exposure. In contrast, no significant effects were observed in the 5 min exposure (Figure 6). The exposure to the vitrification solutions tended to delay embryo development; longer exposure times further delayed embryo development. Although all the hatched larvae were initially able to walk for a while, larvae that hatched following longer exposure times failed to reach the artificial diet and subsequently dried out.

Furthermore, the same experiments were examined with immersion in liquid nitrogen following the application of vitrification solutions (vitrification). Typically, normal embryos exhibit head pigmentation and serosa membrane ingestion 9 days AEL, eventually hatching 11 days AEL. However, a very slow development of the embryos was observed after the treatment of vitrification solutions and vitrification. Then, 11 days AEL (4 days after vitrification), the vitrificated and rewarmed embryos began to show signs of head pigmentation 11 days AEL (4 days after vitrification), with serosa ingestion observed 15 days AEL. As shown in Figure 7, when the embryos at early 1 phase-stage 25 were vitrificated with the EG and PG protocols, the proportions of embryos developing to head pigmentation 11 days AEL increased as the exposure time in vitrification solution was longer. The proportion of embryos developing to head pigmentation was significantly higher (*z* = 2.957, *p* = 0.003) in 20 min exposure than in the 5 min exposure in the EG protocol (Figure 7). Similar results were found in the observation 15 days AEL. There were no embryos developing to serosa ingestion 11 days AEL in either treatment, and only embryos treated with EG vitrification solution for 20 min developed to serosa ingestion 15 days AEL (Figure 7). It should be noted that some of the embryos were damaged during the warming process, particularly during the steps involving Grace’s medium, which resulted in outcomes comparable to those observed in Figure 5.

### 2.5. Sensitivities of Different Embryos Stages to Vitrification

As described in Section 2.4, only embryos vitrified using the EG protocol for 20 min developed to serosa ingestion. We, therefore, used the EG protocol to evaluate the influence of vitrification in embryos at different phases of Stage 25, given that during this stage the chitinization process occurs around 166 h AEL and the permeability of the embryo decreases [15]. Embryos at stage 24 (157 h AEL), early 1 phase-stage 25 (160 h AEL), early 2 phase-stage 25 (163 h AEL), and middle 1 phase-stage 25 (166 h AEL) were treated with the protocol of 2 M EG solution for 30 min followed by vitrification solution of 7.2 M EG with 0.5 M trehalose for 20 min. As shown in Figure 8, the proportions of embryos that develop to head pigmentation increased in later phases of Stage 25 excluding the middle 1 phase-stage 25. During the 15 days AEL, we observed the development of embryos to serosa ingestion at stage 24, early 1 phase-stage 25, and early 2 phase-stage 25. However, they were not present at middle 1 phase-stage 25. The highest proportion (28.07 ± 12.3%) was obtained in early 2 phase-stage 25 (Figure 8).

As shown in Figure 9, the embryos after vitrification and rewarming (Figure 9A) were still alive and developed slowly. The embryos 11 days AEL began to show slight head pigmentation (Figure 9B), and the pigmentation increased 15 days AEL (Figure 9C,D). It should be noted that the embryos with head pigmentation began to exhibit movement from 11 days AEL. Then, 15 days AEL, some larvae ingested their own serosa membranes (Figure 9D). Despite being provided with an artificial diet, they were unable to walk and eventually dried up in hours and died. In addition, the vitrified embryos (Figure 9) that developed to head pigmentation showed a discernible reduction in head pigmentation compared to the permeabilized embryos (Figure 3).

### 2.6. Vitrification Using a Different Strain

Further experimentation was conducted using a diapausing strain (w1) to examine the potential influence of vitrification. The eggs were subjected to an acid treatment 20 h AEL to break the diapause. The dechorionated and permeabilized embryos in early 2 phase-stage 25 were subjected to a 2 M EG solution for 30 min, followed by a vitrification solution of 7.2 M EG with 0.5 M trehalose for 20 min. As a result, the proportion of embryos developing to head pigmentation in the w1 strain was found to be lower than that of the non-diapausing strain pnd-w1 (Figure 8 and Figure 10A). Additionally, no embryos developed to serosa ingestion. As shown in Figure 10B,C, following vitrification treatment, the embryos of the w1 strain still were alive and developed slowly like that of the pnd-w1 strain. However, evidence of tissue damage was observed in w1 embryos at 11 days AEL (Figure 10C).

## 3. Discussion

The success of the cryopreservation of insect embryos depends on their sufficient permeability to CPAs. However, the toxicity and permeation properties of each CPA differ and change by species, developmental stage, concentration, and temperature [6,10,16,17,18]. To date, a limited number of studies have been conducted on the cryopreservation of silkworm embryos (*Bombyx mori*, Lepidoptera: Bombycidae). However, these have not yet yielded successful results [11,19]. In advance of the cryopreservation study, we developed methods to dechorionate and permeabilize silkworm embryos [14,15]. Accordingly, the present study was conducted to investigate the CPA permeability and vitrification as a final step of cryopreservation. To this end, a number of factors were examined, including the types and concentrations of CPAs, exposure time, embryonic stage, and silkworm strains. In line with this objective, the experiments here were basically performed using an embryonic stage of early 1 phase-stage 25, which has been demonstrated to exhibit high permeability and viability [15].

Firstly, we examined the permeability of the silkworm embryos to four CPAs (DMSO, EG, GLY, PG) using the previously developed methods, which enable the precise measurement of relative area changes during exposure to CPAs via a microscope with a camera system [15]. On the view of the permeability, the molecular weight (MW) of CPAs is a significant factor [6]. This is because it can be postulated that a CPA with a smaller MW will permeate the embryos more readily than one with a higher MW. The results of the re-expansion at 120 min of exposure to CPAs, as illustrated in Figure 1, indicated that the permeability of GLY (MW = 92.1) was comparatively lower than that of EG (MW = 62.1), PG (MW = 76.1), and DMSO (MW = 78.1). Ultimately, PG exhibited the highest permeability followed by DMSO, EG, and GLY, which demonstrated the lowest permeability of the four CPAs. However, the permeabilities of PG and DMSO were higher than that of EG, which was incompatible with the MW theory. These findings suggest that alternative mechanisms for the permeability of CPAs may exist in silkworm embryos. It is noteworthy that similar permeability trends to those observed in silkworms for EG, PG, DMSO, and GLY (Figure 2) have been observed in mammalian oocytes [6,20]. In addition, studies on the permeation of EG and GLY through oocytes and embryos of mammals have identified diffusion and facilitated diffusion via aquaporin 3 as key mechanisms [21]. Similar mechanisms such as aquaporins may exist in silkworm embryos and facilitate glycerol transport [22].

Furthermore, according to our findings, following the exposure of embryos to CPAs, a reduction in size was initially observed due to water loss. However, as the CPAs permeated the membranes (i.e., vitelline, serosal), the embryos exhibited gradual re-expansion. Nevertheless, it was observed that the embryos did not regain their initial relative area in any 2 M CPA solution after 120. In the reports on *Drosophila melanogaster* (Diptera: Drosophilidae) [1,10], the embryos in later stages recovered their original shape within 20 to 25 min following exposure to EG. In mammalian embryos, the greatest degree of shrinkage has been observed after seconds of exposure to CPAs [6,20,23]. In comparison, silkworm embryos appear to have a markedly lower permeability to water and CPAs than embryos of mammals and dipteran species. It has been reported that embryo permeability varies considerably between species and developmental stages [6,21,24]. Nevertheless, it is, in fact, hard to succeed in the cryopreservation in silkworm because the eggs are larger/voluminous and possess thicker membranes unlike the mammalian embryos [25,26,27,28,29,30], leading us to conduct further research into these factors.

Secondly, we examined the toxicities of DMSO, EG, GLY, and PG on the silkworm embryos. It is reasonable to assume that the toxicities of CPAs vary between different organisms. Indeed, Luo et al. found that GLY was more toxic than EG and PG in *Spodoptera exigua* (Lepidoptera: Noctuidae) [31], whereas Zhan et al. observed that DMSO was more toxic than EG and PG in *D. melanogaster* [10]. EG is a commonly employed CPA in the cryopreservation of insect embryos [32], with low toxicity in dipteran [8,9,10] and lepidopteran species [31,33]. In contrast, PG is less toxic but has been less frequently employed in studies of *D. melanogaster* [10] and *S. exigua* [31]. In our study, EG and PG were the least toxic of the four CPAs to silkworm embryos. The exposure of silkworm embryos to 2 M of EG and PG for 30 min demonstrated their normal development and the similar results in favorable viability (Figure 3), whereas DMSO and GLY tended to delay the embryonic development (Figure 3B). It is possible that the observed toxicity of DMSO in silkworm embryos may be attributed to its neurotoxic effects [10,34]. On the other hand, although GLY is a CPA, it is a common component of diapausing silkworm eggs [35,36] and has been demonstrated to exert an inhibitory effect on the embryonic development [37]. However, it has been reported that GLY has lower membrane permeability than DMSO [38]. These findings are consistent with our results, which demonstrated that GLY delayed embryonic development (Figure 3B) and exhibited poor permeability to the embryos (Figure 1: as evidenced by the re-expansion), in addition to causing the greatest degree of shrinkage (Figure 2). To prevent the over-rehydration of cells by a rapid influx of water, embryos are often placed in an insect culture medium containing a non-penetrating CPA, such as trehalose [26]. In this study, we also employed trehalose solutions in longer exposure times. Nevertheless, this may not have been sufficient to control the influx of water and remove the CPAs. In accordance with the additional data (Appendix A), the silkworm embryos were more tolerant to lower concentrations of EG and PG (Appendix A), even for longer exposure times. Furthermore, PG seems to be less toxic than EG.

Thirdly, the effects of two vitrification protocols using EG and PG on silkworm embryos were evaluated. The exposure of the embryos to EG- and PG-based vitrification solutions without liquid nitrogen exposure resulted in a reduction in the proportion of viable embryos (Figure 6). This may be attributed to the high concentration of the vitrification solutions, which can lead to various effects such as dehydration [39], osmotic stress [5], and disruptions in the normal metabolism [40]. It was unexpected that longer exposure times to the vitrification solutions resulted in enhanced embryonic viability after vitrification and rewarming (Figure 7). It is possible that due to the slow permeation of EG and PG to silkworm embryos, an insufficient CPA was loaded, and high water contents remained in the embryos, so longer exposure times are required to avoid ice formation.

Fourthly, the impact of the embryonic stage on the viability of silkworm embryos undergoing vitrification was investigated. The selection of the appropriate embryonic stage is a critical aspect for the successful cryopreservation of insect embryos [26]. Embryonic cryopreservation of dipteran embryos has only been attempted at late organogenesis stages, with the most tolerant stage identified as just prior to the development of the embryonic exoskeleton [1,9,10]. Initially, we used embryos at the early 1 phase-stage 25 (160 h AEL), which showed high permeability and viability post permeabilization in preceding studies [15]. However, it should be noted that even subtle changes within an embryonic stage have the potential to greatly affect the success of cryopreservation [8,11,32]. Fukumori et al. reported the highest tolerance to the vitrification in later stages (between Stages 24 and 25) of silkworm embryos [11]. When we examined the different sensitivities within Stage 25 of silkworm embryos, it was found that there were different sensitivities to vitrification and rewarming. The highest tolerance to vitrification was observed at early 2 phase-stage 25 (163 h AEL) (Figure 8), which is just before the composition of chitin (middle 1 phase-stage 25, 166 h AEL) [15]. However, even under the best experimental condition in this study and in line with Fukumori et al.’s observations [11], the silkworm was unable to progress from the 1st to 2nd instar molting stages. This suggests that the same issues required resolution in this study. We observed that embryos exposed to vitrification solution exhibited a reduced degree of head pigmentation (Figure 9) and were unable to walk to the artificial diet after hatching. Consequently, they dried out and died. An insufficient sclerotization of the embryos may result in a rapid loss of body water and subsequent dehydration [11]. This could be attributed to abnormal chitinization/sclerotization, which commences in the epidermis at the same time as the appearance of taenidium.

Fifthly, the vitrification protocol was applied to embryos from diapausing eggs. These embryos developed much more slowly and caused more damage (Figure 9C), in line with a previous study [15], where it was found that the embryos from diapausing eggs had lower water permeability levels than non-diapausing ones. This suggests that embryos from diapausing eggs may also be less permeable to CPAs, which could result in greater embryo damage after vitrification.

The formation of intracellular ice would cause irreversible damage to cells in embryos, resulting in lethal injury [5,41]. Unfortunately, the vitrification protocols in this study were not fully functional, resulting in a very slow embryo development and no embryos reaching the second instar (Figure 7, Figure 8 and Figure 9). To achieve successful cryopreservation, it is essential to consider not only the optimal stage of insect development and the use of specialized freezing devices, but also the balance between intracellular water removal and permeabilization of sufficient CPA in the cell without excess toxicity [1,41]. Despite these considerations, there are still numerous factors to investigate for successful cryopreservation in *B. mori*.

## 4. Materials and Methods

### 4.1. Collection of Eggs and Embryonic Stages

For the experiments, we used two strains: the pnd-w1 and the w1. The pnd-w1 has white-colored eggs with a non-diapause nature, while the w1 has white-colored eggs but with a diapause nature. Newly emerged moths were mated for at least 2 h at 25 °C and then stored for one day at 5 °C; this procedure enable the female moths to lay eggs easily during their collection. To collect the eggs, the females were separated, transferred to a piece of paper, and kept in dark places at 25 °C for one hour. The eggs of the w1 strain, which had reached 20 h AEL, were treated with a hydrochloric acid solution (HCl diluted with distilled water; specific gravity at 15 °C: 1.1100) at 25 °C for 90 min to break the diapause. They were then rinsed in running water for 30 min. Following a 48 h AEL, the eggs were stored in the incubator at 5 °C and subsequently incubated at 25 °C until the defined time in accordance with experiments. We defined the embryonic stages as described in [14,15]: stage 24 (157 h AEL), early 1 phase-stage 25 (160 h AEL), early 2 phase-stage 25 (163 h AEL), and middle 1 phase-stage 25 (166 h AEL).

### 4.2. Dechorionation and Permeabilization

All eggs were attached to a nylon net (30 µm) and dechorionation was performed as described in [14]. They were treated with 30% potassium hydroxide (KOH) for 7 min and 2% sodium hypochlorite (NaClO) for 5 min at 27 °C, rinsed in phosphate-buffered saline (PBS) for 10 min, immersed in 1% sodium carbonate (Na_2_CO_3_) for 1 min, rinsed in PBS for 5 min, and washed in Grace’s insect medium. The permeabilization of the dechorionated eggs was performed as described in [15]; the dechorionated eggs were subjected to the optimized permeabilization treatment of hexane for 30 sec and then transferred in Grace’s insect medium (Gibco Life Technologies, Grand Island, Nueva York, NY, USA) for further 5 min.

### 4.3. Osmotic Response of Embryos to CPAs

The osmotic response of permeabilized eggs to CPAs was evaluated as described in [15]. The permeabilized eggs attached to a nylon net were taken out of Grace’s medium and placed and clamped to a tissue culture dish. Next, 2 mL of Grace’s medium was added to the dish, incubated for 5 min, and placed on the stereomicroscope (SZX16; Olympus Co., Shinjuku City, Japan) equipped with a digital camera (DP71). Grace’s medium was removed with a pipette, and 2 mL of DMSO, EG, PG (Wako, Japan), and GLY (Invitrogen, Carlsbad, CA, USA) solutions with different concentrations (0.5, 1, 2, and 4 M) in Grace’s medium were then added. The embryos were imaged digitally during CPA exposure using a time-lapse system at 5 min intervals over a 120 min period. The imaging was conducted using the cellSens Standard software version 1.14 (Olympus Co., Japan); all this process was performed at room temperature (25 °C) controlled using an air conditioner. The area of each digitalized egg was measured by a computer using ImageJ software version 1.53h (NIH, Bethesda, MD, USA; https://imagej.nih.gov/ij/ (accessed on 5 December 2023)). The area of the shrunken eggs was defined by the contour of the membranes (vitellin and serosal), and then the shrinkage changes against the initial area in Grace’s medium were represented as the “relative area”. Only embryos with similar shapes to that of an ellipse were analyzed. The embryos with significant morphological alterations, damaged, or not showing a typical shrink response were excluded from the analysis.

### 4.4. Treatment of CPAs for Assessment of Toxicity

The permeabilized eggs attached to a nylon net were exposed to solutions of DMSO, EG, PG (Wako, Japan), and GLY (Invitrogen, USA) in Grace’s medium, with different concentrations (1 and 2 M) and exposure times (30, 60, 90, and 120 min). Next, the embryos were blotted on a sterilized filter paper to remove the excess CPA solution and then rinsed in Grace’s medium containing a gradually reduced concentration (0.5 M for 5 min; 0.25 M for 10 min; 0.125 M for 10 min) of trehalose (Wako, Japan) to control osmotic pressure and remove the CPAs. The embryos were blotted on a sterilized filter paper to remove as much trehalose solution as possible, and then washed in Grace’s medium twice for 5 min and cultured in the dry–moist system described in [14]. All these steps were performed at room temperature (25 °C), which was controlled using an air conditioner.

### 4.5. Vitrification and Rewarming of Embryos

The vitrification and rewarming of embryos were performed as shown in Figure 11. The permeabilized eggs attached to a nylon net were exposed to 2 M EG or PG solutions in Grace’s medium for 30 min at room temperature (25 °C), which was controlled using an air conditioner. The embryos were blotted on a sterilized filter paper to remove the excess CPA solution and were dehydrated by exposure to a vitrification solution containing 7.2 M EG or 5.5 M PG in Grace’s medium with 0.5 M trehalose at 4°C in a refrigerator (MPR-414FS, Medicool, SANYO, Gunma, Japan) for 5, 10, or 20 min. Next, the embryos were rapidly blotted on a sterilized filter paper to remove the majority of the vitrification solution and then rapidly plunged into liquid nitrogen. The embryos were maintained in liquid nitrogen for 15 to 20 min (Figure 11A). For rewarming, the embryos were exposed to nitrogen vapor for 1 min [8] (around 1 cm above liquid nitrogen). Following this, the embryos were rapidly submerged in 5 mL of 0.5 M trehalose solution in Grace’s medium at 40 °C for 5 min. Next, the embryos were transferred to 0.25 M and 0.125 M trehalose in Grace’s medium for 10 min each. Finally, the embryos were blotted on a sterilized filter paper to remove the majority of the trehalose solution, washed in Grace’s medium twice for 5 min (Figure 11B), and cultured in the dry–moist system as described above. All steps were performed at room temperature (25 °C) controlled using an air conditioner.

### 4.6. Assessment of Development and Viability

The assessment of the CPA toxicity on development and viability was performed using a stereomicroscope (LED300SLI, Leica Microsystems, Wetzlar, Germany). Embryos that exhibited adequate developmental progress 9 days AEL were scored as “developing embryos”. Larvae that completed the ingestion of the membranes enveloping embryo, showed the forming head and body pigmentation, and walking 11 days AEL were scored as “hatching”. The number of “embryos developing to the second instar” in each treatment was recorded 19 days AEL. Additionally, the assessment of development and viability after vitrification was performed 11 and 15 days AEL. Embryos with head pigmentation were scored as “head pigmentation”, while embryos with the ingestion of the membranes were scored as “serosa ingestion”.

### 4.7. Statistical Analysis

All statistical analyses were performed using SPSS software (Version 27.0. Armonk, NY, USA: IBM Corp.). Multiple analyses were performed using Kruskal–Wallis test and post hoc Dunn’s test. CPA and dehydration toxicity were compared to untreated permeabilized embryos (control). Vitrification data were compared among each experimental group. Significant variation was defined as *p* < 0.05.

## 5. Conclusions

We investigated the permeabilization and toxicity of four types of CPAs with the objective of enhancing the cryopreservation in silkworm, *B. mori*. The embryos initially shrank and gradually re-expanded during the exposure to CPAs. The greatest degree of shrinkage in all the CPAs was observed at around 30 min after treatment, with high survival rates after culturing. However, some embryos with abnormal shapes were observed, despite EG and PG being the least toxic of the four CPAs to silkworm embryos. Additionally, even embryos with a normal shape after vitrification could not survive to second instar larvae. To achieve successful cryopreservation in the silkworm, it may be necessary to conduct more detailed studies of additional factors.

## Figures and Tables

**Figure 1 ijms-25-11396-f001:**
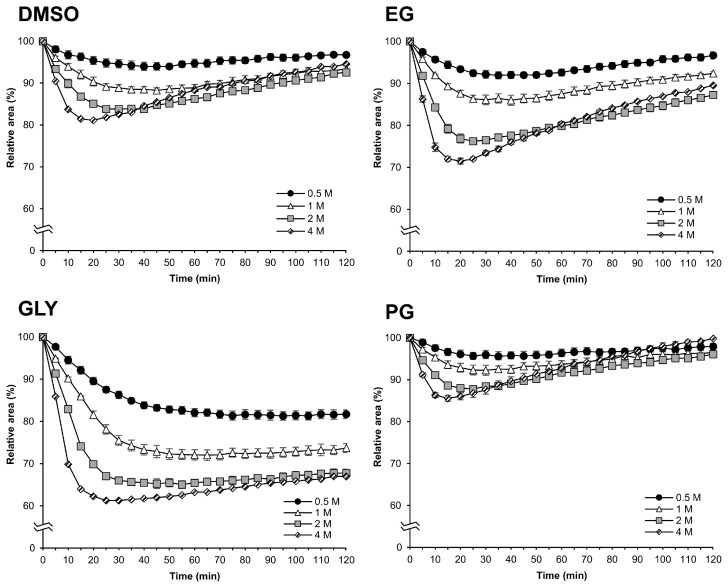
The relative area of embryos at the early 1 phase-stage 25 (appearance of taenidium, 160 h AEL) during exposure to DMSO, EG, GLY, and PG solutions and different concentrations. The permeabilized embryos of the pnd-w1 strain were exposed to 0.5, 1, 2, and 4 M of CPAs for 120 min at 25 °C. The experiment was repeated three times (*n* = 3); 9–12 eggs were used in each replicate experiment. Symbol marks and error bars represent the mean ± SE.

**Figure 2 ijms-25-11396-f002:**
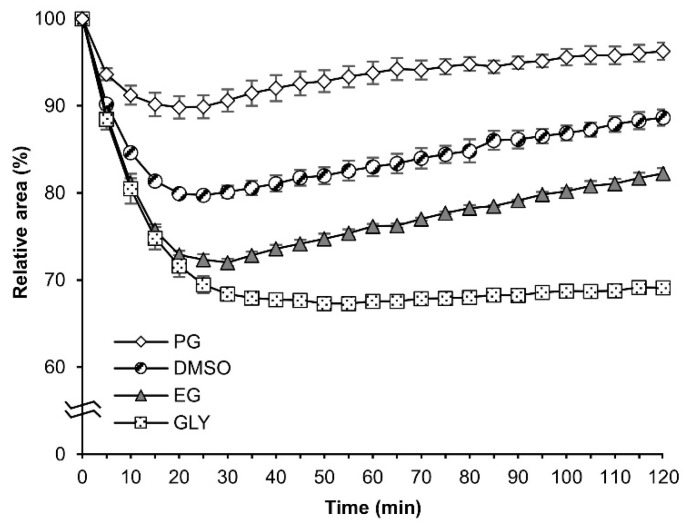
The relative area of embryos at the early 1 phase-stage 25 (appearance of taenidium, 160 h AEL) during exposure to 2 M DMSO, EG, GLY, and PG solutions. The permeabilized embryos of the pnd-w1 strain were exposed to CPAs for 120 min at 25 °C. The experiment was repeated three times (*n* = 3); 10–12 eggs were used in each replicate experiment. Symbol marks and error bars represent the mean ± SE.

**Figure 3 ijms-25-11396-f003:**
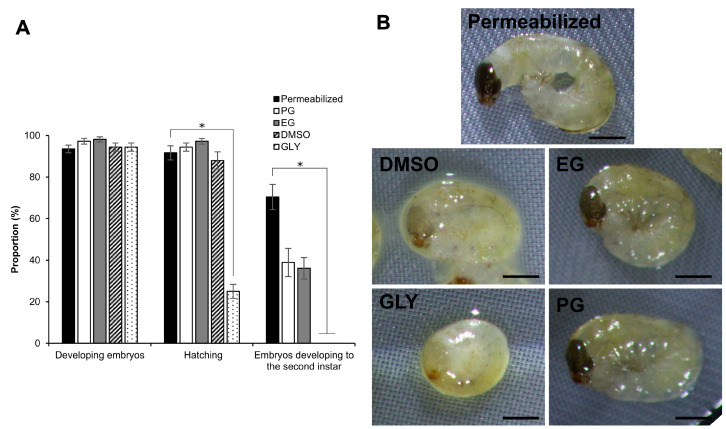
Effect of different 2 M CPAs on embryos at the early 1 phase-stage 25 (appearance of taenidium; 160 h AEL). (**A**) The permeabilized embryos of the pnd-w1 strain were exposed to 2 M DMSO, EG, GLY, and PG solutions for 30 min at 25 °C. *Permeabilized* represents no exposure to CPA solution. The experiment was repeated nine times (*n* = 9); 12 embryos were used in each replicate experiment. Bars and error bars represent the mean ± SE. * *p* < 0.05 (Kruskal–Wallis test and post hoc Dunn’s test). (**B**) Images of embryos 2 days (216 h AEL) after exposure to 2 M CPAs for 30 min. Scale bar: 500 µm.

**Figure 4 ijms-25-11396-f004:**
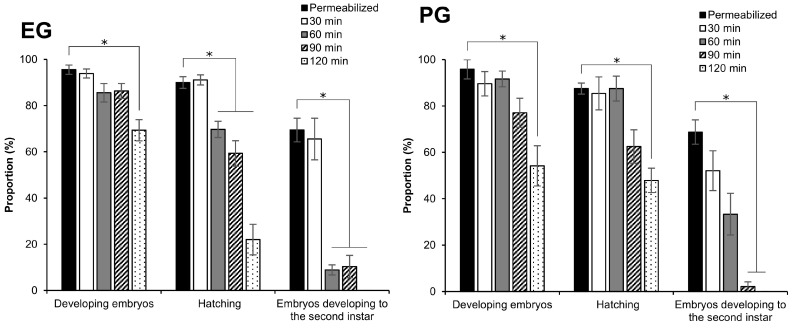
Effect of exposure time in 2 M EG and PG solutions on embryos at the early 1 phase-Stage 25 (appearance of taenidium; 160 h AEL). The permeabilized embryos of the pnd-w1 strain were exposed to 2 M EG and PG solutions for 30, 60, 90, and 120 min at 25 °C. Permeabilized represents no exposure to CPAs. The experiment was repeated four to six times (*n* = 4–6); 10–12 embryos were used in each replicate experiment. Bars and error bars represent the mean ± SE. * *p* < 0.05 (Kruskal–Wallis test and post hoc Dunn’s test).

**Figure 5 ijms-25-11396-f005:**
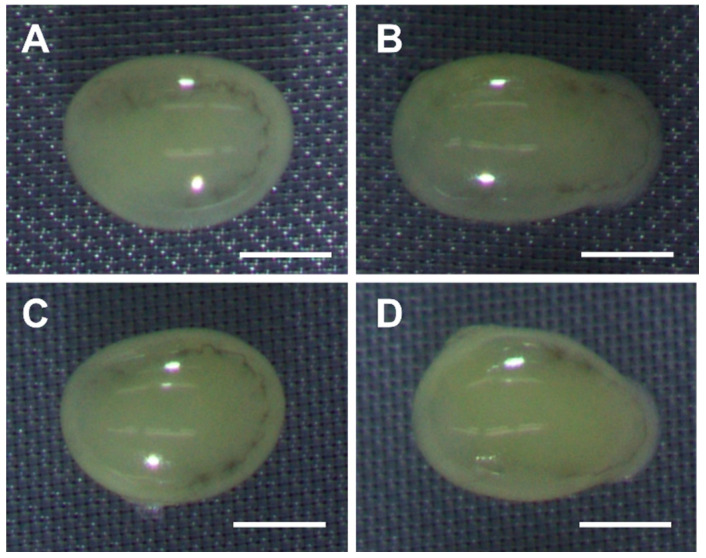
Images of embryos at the early 1 phase-stage 25 (appearance of taenidium; 160 h AEL) of the pnd-w1 strain after exposure to CPAs. The permeabilized embryos were exposed to 2 M EG and PG solutions. (**A**) No damaged embryo after 60 min of exposure to EG solution. (**B**) Damaged embryo after 60 min of exposure to EG solution. (**C**) No damaged embryo after 90 min of exposure to PG solution. (**D**) Damaged embryo after 90 min of exposure to PG solution. Scale bar: 500 µm.

**Figure 6 ijms-25-11396-f006:**
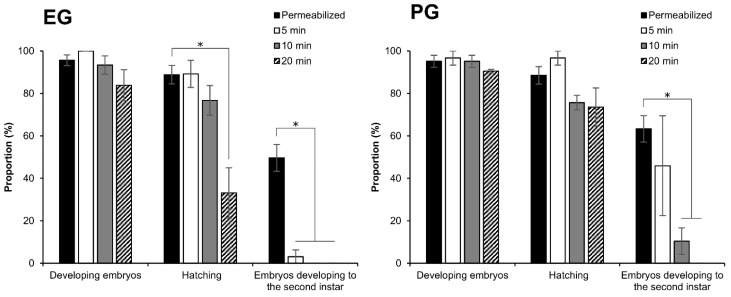
Effect of exposure time in vitrification solutions on embryos at the early 1 phase-stage 25 (appearance of taenidium; 160 h AEL) without immersion in liquid nitrogen. The permeabilized embryos of the pnd-w1 strain were exposed to 2 M EG for 30 min followed by 7.2 M EG solution with 0.5 M trehalose for 5, 10, and 20 min or 2 M PG for 30 min followed by 5.5 M PG solution with 0.5 M trehalose for 5, 10, and 20 min. Permeabilized represents no exposure to CPAs. The experiment was repeated three to four times (*n* = 3–4); 10–12 embryos were used in each replicate experiment. Bars and error bars represent the mean ± SE. * *p* < 0.05 (Kruskal–Wallis test and post hoc Dunn’s test).

**Figure 7 ijms-25-11396-f007:**
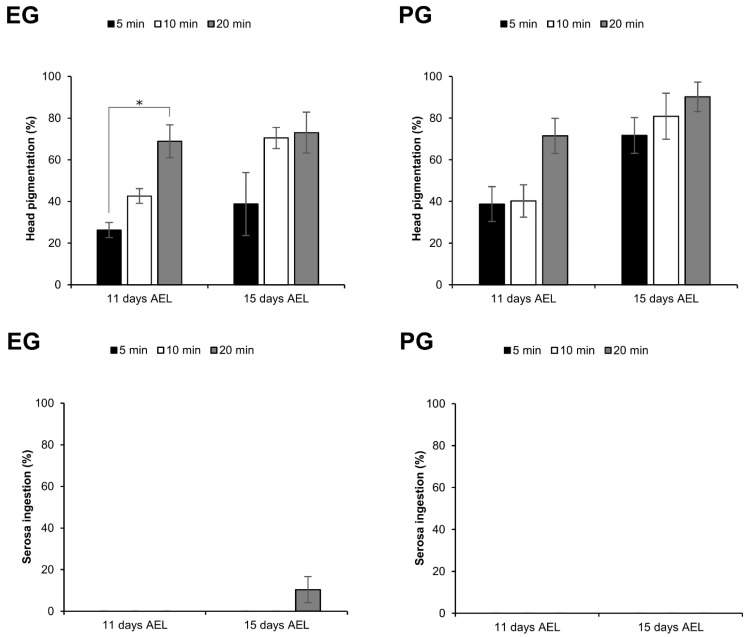
Effect of immersion in liquid nitrogen (vitrification) after different exposure times to vitrification solutions on embryos at the early 1 phase-stage 25 (appearance of taenidium; 160 h AEL). The permeabilized embryos of the pnd-w1 strain were exposed to 2 M EG for 30 min followed by 7.2 M EG solution with 0.5 M trehalose for 5, 10, and 20 min or 2 M PG for 30 min followed by 5.5 M PG solution with 0.5 M trehalose for 5, 10, and 20 min. Next, the embryos were plunged in liquid nitrogen. The experiment was repeated four times (*n* = 4); 10–12 embryos were used in each replicate experiment. Bars and error bars represent the mean ± SE. * *p* < 0.05 (Kruskal–Wallis test and post hoc Dunn’s test).

**Figure 8 ijms-25-11396-f008:**
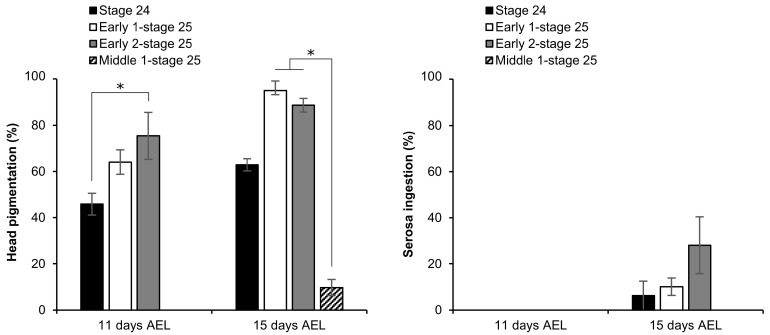
Effect of vitrification on embryos at different phases of Stage 25 (appearance of taenidium). The permeabilized embryos of the pnd-w1 strain were exposed to 2 M EG for 30 min followed by 7.2 M EG solution with 0.5 M trehalose for 20 min and immersed in liquid nitrogen; Stage 24, 157 h AEL; Early 1 phase-stage 25, 160 h AEL; Early 2 phase-stage 25, 163 h AEL; Middle 1 phase-stage 25, 166 h AEL. The experiment was repeated four times (*n* = 4); 9–12 embryos were used in each replicate experiment. Bars and error bars represent the mean ± SE. * *p* < 0.05 (Kruskal–Wallis test and post hoc Dunn’s test).

**Figure 9 ijms-25-11396-f009:**
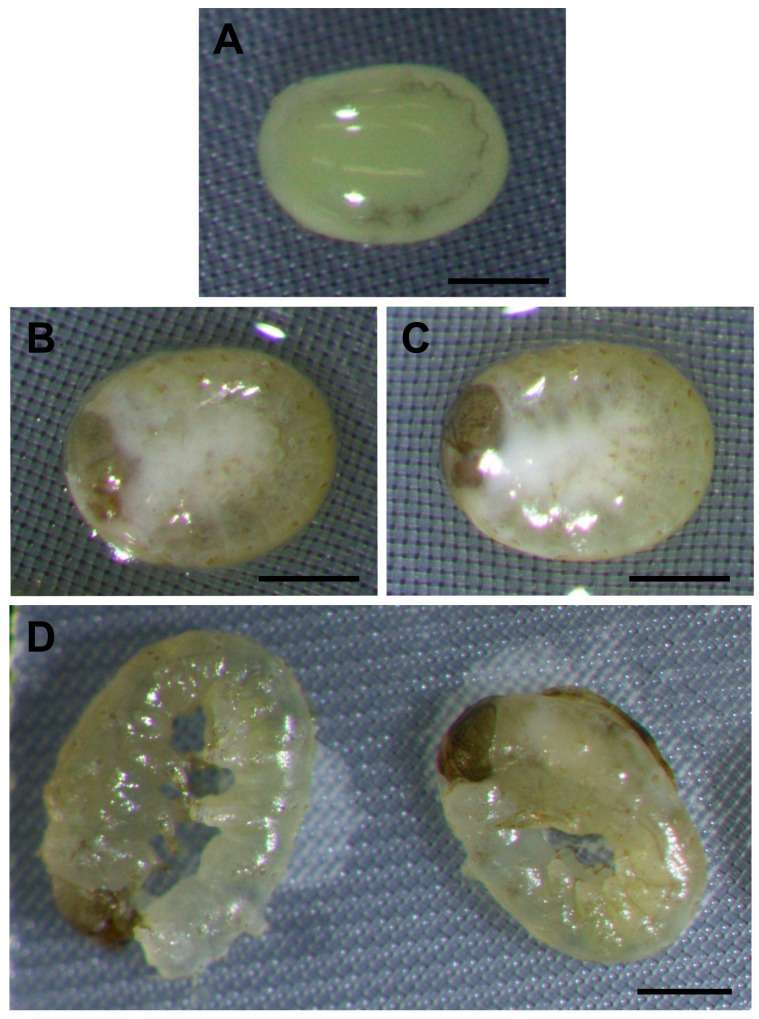
Development after vitrification and rewarming using early 2 phase-stage 25 (appearance of taenidium; 163 h AEL) embryos of the pnd-w1 strain. The permeabilized embryos were exposed to 2 M EG for 30 min followed by 20 min immersion in a 7.2 M EG solution with 0.5 M trehalose and then plunged into liquid nitrogen. (**A**) Embryo immediately after rewarming. (**B**) Embryo 4 days after rewarming (11 days AEL). (**C**) Embryo without serosa ingestion 8 days after rewarming (15 days AEL). (**D**) Embryos with serosa ingestion 8 days after rewarming (15 days AEL). Scale bar: 500 µm.

**Figure 10 ijms-25-11396-f010:**
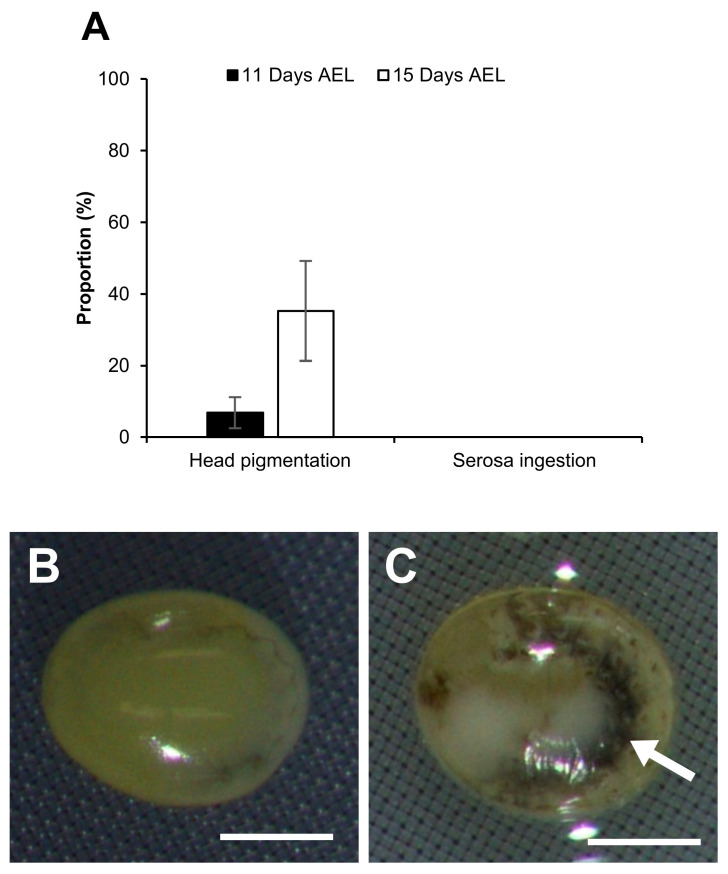
Effect of vitrification on embryos at the early 2 phase-stage 25 (appearance of taenidium; 163 h AEL) of the diapausing strain w1. The permeabilized embryos were exposed to 2 M EG for 30 min followed by 7.2 M EG solution with 0.5 M trehalose for 20 min and immersed in liquid nitrogen. (**A**) Proportion of embryos developing to head pigmentation and embryos developing to serosa ingestion. The experiment was repeated three times (*n* = 3); 11–12 embryos were used in each replicate experiment. Bars and error bars represent the mean ± SE. (**B**) Embryo immediately after warming. (**C**) Embryo 4 days after warming (11 days AEL). White arrow indicates damaged tissue. Scale bar: 500 µm.

**Figure 11 ijms-25-11396-f011:**
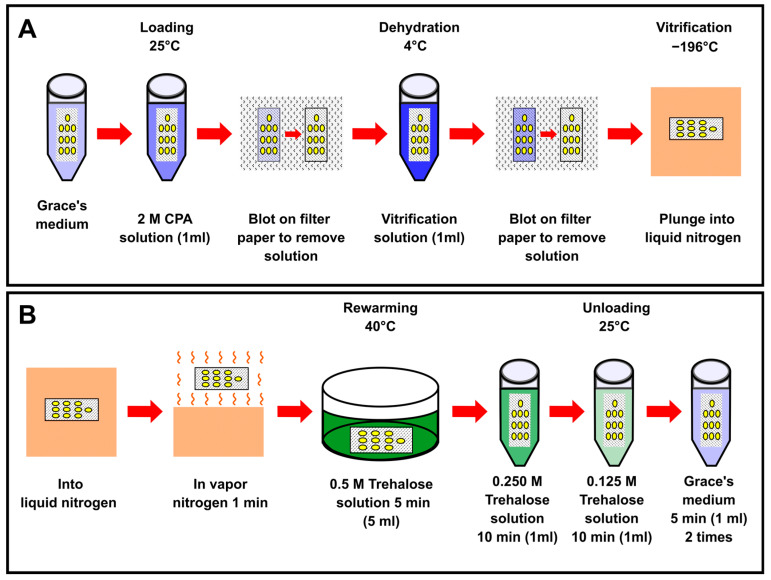
Protocol for cryopreservation of silkworm embryos. (**A**) Outline of vitrification protocol. The permeabilized embryos attached to a nylon net were immersed in 2 M CPA solution at 25 °C, blotted on filter paper to remove the excess solution, transferred to the vitrification solution at 4 °C, blotted on filter paper to remove the excess solution, and finally plunged into liquid nitrogen. The composition of the CPA and vitrification solutions varied in the defined experiments. (**B**) Outline of rewarming. The vitrificated embryos were exposed to vapor nitrogen for 1 min (around 1 cm above liquid nitrogen), then rewarmed in 0.5 M trehalose solution at 40 °C, followed by 0.25 M and 0.125 M trehalose solutions. Finally, the embryos were immersed twice in Grace’s medium.

## Data Availability

Data are contained within the article and Appendix A.

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
