# Peer review of "Permeability and Toxicity of Cryoprotective Agents in Silkworm Embryos: Impact on Cryopreservation"

_ijms, 2024, doi:10.3390/ijms252111396_

Round 1
Reviewer 1 Report
Comments and Suggestions for Authors
The authors have studied the osmosis and toxicity of four cryoprotective agents (CPAs) as well as the effects of vitrification on the viability of silkworm embryos. Also, the authors investigated the potential factors, such as the types and the concentration of CPAs, exposure time, embryonic stage, and silkworm strains, affecting the viability of embryos cryopreserved. This is a piece of nice work. However, the manuscript needs being polished before be accepted as a publication of the Journal, IJMS.
Minor points:
1. ‘……examined through measuring and analyzing the relative area changes…” what’s meaning of the relative area? Here, it is better to examine the changes of embryos’ volume or diameter.
2. Some descriptions are inaccurate, such as, the sentence in line 68-69, line 113-114, etc. The authors should provide professional descriptions but not the colloquial language
3 There are many long sentences, being hard for reading and understanding. For example, the sentence in line 202-206.
Comments on the Quality of English LanguageThe English writing needs to be improved.
Reviewer 2 Report
Comments and Suggestions for Authors
looks good as far as i can judge (i did not recalculate the numbers)
please just edit the english a bit, for example:
+++
line 23:
"We found that over 21% of embryos treated with EG at the early 2 phase of stage 25: 163 h after egg laying (AEL), developed and progressed to serosa"
no comma
+++
line 58:
"The results obtained here would be valuable for the development of embryo cryopreservation protocols for other insects as well."
why "would"? will? could? are valuable?
+++
line 100:
"The embryos treated by EG and PG developed to the second instar although observing a decrease"
we observed / was observed
+++
line 103:
"tended to delay the development of the embryos; more remarkable in GLY"
more remarkably or this was more remarkable
+++
line 108
"Permeabilized represents no exposure to CPA solution."
please write Permeabilzed in italics so it is clear you refer to the legend of the table
+++
line 162:
"However, the hatching was significantly reduced in the exposure of 20 min (z = 2.742, p = 0.006) and their growth to the second instar dramatically reduced in all exposure times (Figure 6)."
in? was.... reduced in?
+++
line 172
"larvae derived from longer exposure times did not reach the artificial"
is "derived" a technical lab slang term here? you mean "hatched after"?
+++
line 256
"As shown in the Figure 10B and C,"
as shown in fig. 10B
+++
line 313
"Second, we examined the toxicities of DMSO, EG"
Secondly
+++
line 414
"The imaging of the embryos during the exposure to CPAs was digitalized"
i do not understand: what do you mean by "the imaging was digitalized"? the images where stored digitally?
etc. — maybe a native english speaker could check the few english language issues
Comments on the Quality of English Languageit is fine except minor issues that should be resolved by an english language editor "on the go"
Reviewer 3 Report
Comments and Suggestions for Authors
While there is plenty of room for improvement and fine tuning this study, I believe that this study lays the better foundations of germplasm storage technology for silkworm embryos compared to a few other studies from the past. While the lepidopteran embryos in general are very recalcitrant to permeabilization due to their formidable extra embryonic membranes and especially the chorion, the authors’ previous study in 2023 (Bioengineering) resolved the majority of this problem. There are a few surprises that were unexpected including the DMSO toxicity. It appears that the more permeable the CPA is, the loss of embryos is higher. Overall, this is very valuable research and therefore I would like to commend the authors.
I have a few suggestions enumerated below.
1) I would like to request the authors to re-read the introduction and discussion just to make some sentences more legible.
2) I would encourage the authors to add a diagrammatic note on the differences between stages 24-26. It would immensely help other hoping to repeat this study in state selection which is a critical aspect of insect embryo cryopreservation. I am more used to referring to the excellent compendium by an authority in the field of insect embryology, Dr. Keiichiro Miya entitled, “The early embryonic development of Bombyx Mori - An ultrastructural point of view” (2003) and published by Gendaitosho. Herein, I believe the stage 25 that the authors refer to, and when the taenidium is apparent, is equivalent to the stage 11 noted by Dr. Miya. However, the authors do use a precedent for this staging method from Fukomori et al. (2020 Cryobiology Vo. 95). This request of mine to elaborate and diagrammatically represent the developmental stages garners even more importance when the readers would try to decipher what the authors are referring to in Figure 8 / section 2.5.
3) Figure 5 shows undamaged and damaged embryos after 2M EG and PG treatments for 60 and 90 minutes respectively. I am not sure about the significance of this figure. Is it just to indicate that there could be both undamaged as well as damaged embryos?
In addition, some extremely minor changes that I would like suggest include,
Line 29 - change to: “… during cryopreservation is dependent on…”
Line 34: change to: “...because a critical amount of the CPA must permeate the cell for successful…”
Line 44: change “the CPA kind” to “the class or type of CPA”
Line 46-47: Sentence has a font size change Line 276: “We have already developed methods to dechorionate and permeabilize silkworm embryos”.
Lines 296-99: These sentences must be reformatted for better readability.
Line 303: In the reports on Drosophila…
Line 305: In mammalian embryos…
Line 311: … the eggs are larger/voluminous and possess thicker membranes unlike the mammalian embryos leading us further research in these factors.
Line 313: …on the silk worm embryos…
Line 338: …exposure of the embryos to EG and PG based vitrification solutions without liquid nitrogen exposure…
Further on: Please reformat the sentences with assistance from an English speaker. However, I believe the authors have conveyed the results well.
Comments on the Quality of English LanguageEvery sentence in both the introduction as well as the discussion must be read carefully and reformatted for readability.
